# Tertiary Lymphoid Structures: A Potential Biomarker for Anti-Cancer Therapy

**DOI:** 10.3390/cancers14235968

**Published:** 2022-12-02

**Authors:** Ji’an Zou, Yingzhe Zhang, Yue Zeng, Yurong Peng, Junqi Liu, Chaoyue Xiao, Fang Wu

**Affiliations:** 1Department of Oncology, The Second Xiangya Hospital, Central South University, Changsha 410011, China; 2Hunan Cancer Mega-Data Intelligent Application and Engineering Research Centre, Changsha 410011, China; 3Hunan Key Laboratory of Tumor Models and Individualized Medicine, The Second Xiangya Hospital, Central South University, Changsha 410011, China; 4Hunan Key Laboratory of Early Diagnosis and Precision Therapy in Lung Cancer, The Second Xiangya Hospital, Central South University, Changsha 410011, China

**Keywords:** tertiary lymphoid structures, immunotherapy, chemotherapy, biomarker, cancer

## Abstract

**Simple Summary:**

The tumor immune microenvironment is an important area of tumor research. Tertiary lymphoid structures, as a specific component of this, have recently begun to attract the attention of researchers and clinicians. This review describes the structure and composition of tertiary lymphoid structures and demonstrates the current findings related to tertiary lymphoid structures in clinical studies. The authors aim to deepen the understanding of tertiary lymphoid structures and to demonstrate their potential in tumor treatment strategies and prognostic assessment.

**Abstract:**

A tertiary lymphoid structure (TLS) is a special component in the immune microenvironment that is mainly composed of tumor-infiltrating lymphocytes (TILs), including T cells, B cells, DC cells, and high endothelial venules (HEVs). For cancer patients, evaluation of the immune microenvironment has a predictive effect on tumor biological behavior, treatment methods, and prognosis. As a result, TLSs have begun to attract the attention of researchers as a new potential biomarker. However, the composition and mechanisms of TLSs are still unclear, and clinical detection methods are still being explored. Although some meaningful results have been obtained in clinical trials, there is still a long way to go before such methods can be applied in clinical practice. However, we believe that with the continuous progress of basic research and clinical trials, TLS detection and related treatment can benefit more and more patients. In this review, we generalize the definition and composition of TLSs, summarize clinical trials involving TLSs according to treatment methods, and describe possible methods of inducing TLS formation.

## 1. Introduction

At present, with the continuous research on anti-tumor immunity, immunotherapy–especially with respect to immune checkpoint inhibitors led by anti-PD-1/PD-L1–has become a widely accepted therapy in clinical practice [1]. At this stage, immunotherapy targeting immune cells, particularly that which focuses on T cells, has emerged as a powerful weapon against tumors, and includes immune checkpoint blockades, adoptive cellular therapies, and cancer vaccines [2]. However, a large number of cancer patients do not benefit from this novel and effective treatment. This is mainly due to the complexity and heterogeneity of the tumor microenvironment (TME) and the diversity of the immunomodulatory network [3]. The intensity of PD-L1 expression on tumor cells and the mutational burden of the tumor have been considered as biomarkers for determining the population that is effective in immunotherapy. In addition, tumor-infiltrating lymphocytes (TILs) have been shown to be highly correlated with efficacy, but they have not been used as a predictive biomarker for patient selection [4,5]. Although the efficacy of immunotherapy combined with other treatments in populations without PD-L1 expression has been demonstrated in previous studies [6], the mechanisms need to be explored further. Some studies have shown that tertiary lymphoid structures (TLSs)–defined by clusters of immune-infiltrating cells–in tumors or the tumor periphery are mostly correlated with better prognosis in patients, whether or not PD-L1 is expressed [7]. A TLS is a spatial structure composed of B cells in the center, surrounded by T cells and a variety of immune cells and immune-related cells, with no envelope covering its surface. Similar to SLOs, TLSs regulate the immune microenvironment by recruiting circulating immune cells and enhancing local immune action. As sites for the generation of circulating immune cells that control tumor progression and better prognosis, independent of the expression of PD-L1, TLSs hold great prospects and potential. In order to better understand what kind of TLSs may benefit patients and the role of TLSs in tumor immunotherapy, in this paper we summarize the overall roles of TLSs and their components, as well as clinical trials using TLSs as markers and related research in recent years. 

## 2. The Function of TLSs as Complete Structures

Tertiary lymphoid structures, which are also known as ectopic lymphoid organs, are aggregates of immune cells that appear in tissues where no secondary lymphoid organ (SLO) exists. According to a large number of studies in the past year, TLSs are usually observed at inflammatory sites in response to autoimmune diseases, infectious diseases, organ transplantation, inflammatory disorders, and tumors. There is mounting evidence suggesting their formation is closely related to immune responses mediated by exposure to chronic inflammation. TLSs usually lead to a persistent autoimmune response with a negative impact on subsequent pathological conditions in other diseases [8,9,10]. However, TLSs are still considered to be a favorable prognostic and predictive factor in tumors [11,12,13,14,15,16]. TLSs are able to provide local anti-tumor immunity, independent of SLOs, which has been demonstrated in the absence of SLOs in mouse models [17].

At present, TLSs can be detected by methods including H&E, immunohistochemistry (IHC), multiplex immunofluorescence (IF) techniques, various chemokines/cytokines of TLS detection, and related gene expression [18]; however, there is no TLS detection and counting criterion that is widely accepted in clinical practice. On H&E-stained slides, TLSs were identified morphologically as distinct ovoid lymphocytic aggregates presenting HEV and/or a germinal center [19]. A TLS has been described in detail as a germinal center consisting of CD20+B cells and plasma cells, surrounded by CD3+T cell zones with DCs and peripheric HEVs, resembling SLOs in formation, structure, and function [18,20]. Therefore, SLOs are the most suitable model by which to better understand how TLSs are formed; SLOs have been proven to have many similarities with TLSs in preclinical mouse models [21]. The genesis of TLSs results from a highly ordered sequence of events involving interactions between hematopoietic and non-lymphoid stromal cells, in which cytokines, chemokines, adhesion molecules, and survival factors including CXCL13, CXCL12, CCL19, CCL21, the tumor necrosis factor superfamily, IL-6, and IL-7 [22,23] play key roles as molecular components [24]. Additionally, one study recently demonstrated that TGF-β-mediated silencing of STAB-1 that induces Tfh cell differentiation and the subsequent formation of intra-tumor TLSs further sheds light on the origin of TLSs in complex TME [25]. 

TLS density, location, and maturation have been demonstrated to have close associations with favorable clinical outcomes [19,26,27,28,29,30], including disease-free survival (DFS) and overall survival (OS). TLS density is correlated with germinal center formation and the expression of genes associated with adaptive immune response, and it is a strongly independent prognostic marker in lung cancer, colorectal cancer, pancreatic cancer, and breast cancer [31,32,33,34]. There have also been opposing observations that TLS density has no or limited relation with better survival [35,36]. In patients with lung squamous cell carcinoma who received neoadjuvant chemotherapy, the density of TLSs in tumors became similar, losing its predictive effect [27]. The high TLS density in tumors is usually associated with high TIL density as well as with the expression of PD-1/PD-L1, suggesting the potential benefit from immunotherapies [37,38]. However, it was found that a subtype of glioma with high immune infiltrations and TLSs resulted in poor prognosis. This may be because the beneficial effect of TLSs could be reduced in patients with higher immunosuppressive cells such as myeloid-derived suppressor cells (MSDC) [39]. With regard to TLS location, many studies have reported the persistence of TLSs in tumors or peritumors. Peritumoral TLSs often play a favorable role in prognosis in lung cancer [27], pancreatic cancers [40], CRC [41], oral squamous cell carcinoma [42], metastases of melanoma [43], and ovarian cancer [44], but they are also associated with poor prognosis and invasive metastasis in breast cancer [19]. Intra-tumoral TLSs are described as a favorable maker in lung cancer [45] and HCC [46]. With regard to TLS maturation, according to the works of Silina et al., the development of tumor-associated TLSs follows sequential stages of maturation: (1) early TLSs (E-TLSs), T cells, B cells, and CXCL13-expressing perivascular cells gather into clusters without FDC; (2) primary follicle-like TLSs (PFL-TLSs), i.e., TLSs containing FDC without GCs; 3) secondary follicle-like TLSs (SFL-TLSs), i.e., TLSs are analogous to the SLO follicles [27]. E-TLSs lacking germinal centers may favor immune evasion and progression to full-blown HCC in liver precancerous lesions. The presence of mature TLSs is also associated with an improved objective response rate, progression-free survival, and overall survival independently of PD-L1 expression status and CD8+ T-cell density [7]. In general, TLSs vary widely in density, location, maturation, and components, as well as in their proportions in different individuals, and this high heterogeneity needs better classification and further research for application in clinical practice.

As mentioned above, although TLS plays a positive role in anti-tumors immunity in the vast majority of reports, there are still some studies that report its negative effects. Peri-cancerous TLS was considered a major contributor to adverse effects on prognosis [19]. One study showed the abundance of Treg cells in intra-tumoral TLS increased significantly with the increase in peri-cancerous TLS regions [47]. This suggested that Treg cells in peri-cancerous TLS may be responsive to suppression of the anti-tumor response and the interplay between peri-cancerous and intra-tumoral immune cells. An additional prognostic disadvantage was immature TLS. In immature TLS, the types and numbers of immune cells vary greatly, such as B cells, which were low and produced immunosuppressive cytokines in immature TLS [48]. In addition, TLS was found to be associated with immune-related adverse events [49].

## 3. The Role of TLS Components in Tumor-Specific Immune Response

TLSs, as important factors associated with a series of anti-tumor specific immune responses, play multiply significant roles in tumor progression and suppression in TME. Earlier research has suggested that the double roles of each TLS might differ depending on their composition [50]. The cellular components of TLSs affect the function of the anti-tumor immune response in different types of cancer. Therefore, to further understand the prognostic value of TLSs and their dual role in anti-tumor immunity, differences in TLS components and their ratios need to be considered. Next, we focus on five major components of TLSs: T cells, B cells, DCs, HEVs, and TLS-associated cells (Figure 1). 

### 3.1. T Cells

Naïve CD3+T cells recruited by TLSs become activated, proliferate, and differentiate based on the local tumor antigen presentation, cytokine milieu, and expression of costimulatory molecules, to result in several subsets of effector CD4+ T helper cells (Th), effector CD8+ T cytotoxic cells (CTL), and a small amount of memory T cells (Tm). The subtypes of Th cells can activate anti-tumor immunity directly or stimulate T cytotoxic cells to activate anti-tumor immunity, and some specific species can also inhibit immune cells from activity. Resembling SLOs, different subtypes of Th cells and their secreting cytokines, as well as chemokines, have mutual inhibition competition in TLSs. According to digital spatial-profiling data, T cells in tumors without TLSs had a dysfunctional molecular phenotype, which suggests that TLSs play a key role in the immune microenvironment by conferring distinct T cell phenotypes [15]. 

CD4+Th-1 cells are characterized by T-bet and production of IL-2, interferon γ (IFNγ), and so on. IFN-γ is a pleiotropic cytokine that plays an important role in anti-tumor immunity by directly mediating tumor rejection and recruiting and activating innate and adaptive immune cells in TME. IL-2, which promotes T cell proliferation and maintains its functional activity, has been used in patients with metastatic melanoma and kidney cancer. The production of these cytokines by Th-1 cells is crucial to anti-tumor immunity mediated by CD8+ T cells. However, interestingly, a previous study showed that high infiltration in Th-1 cells and high numbers of CD20+ B-cell follicles–both of them usually aggregating with structures considered as TLSs–were associated with better relapse-free survival in gastric cancer [51]. The densities of Th-1 cells and T follicular helper cells (Tfh) are both reported to be positively correlated with overall survival (OS) in nasopharyngeal carcinoma [52], and the latter are vital to B cells during germinal center (GC)-reactions in SLO [53]. This means that TLSs may exert an anti-tumor immune function through allowing T cell and B cell coordination. Although there is increasing evidence confirming the importance of humoral immunity in TLSs, a high ratio of Th-2 cells in TLSs, which are regarded as promoters of humoral immunity, was identified as a remarkably independent risk factor for recurrence in CRC, and the ratio increased in metastatic tumors in previous studies [50]. Although direct evidence that Th-2 cells can suppress anti-tumor immunity and promote tumor progression is lacking for TLSs, findings regarding the TME suggest that Th2 cells can produce IL-4 and IL-13, with the former increasing the expression of epidermal growth factor to enhance neoplastic extravasating into the circulation, and the latter inhibiting the CD8+ cytotoxic T cell (CTL) response indirectly by increasing TGF-β production by myeloid cells in the tumor [54,55].

Tfh cells are usually the dominant subpopulation and are characterized by high expression of CXCR5, PD-1, ICOS, BCL-6, and CD40, as well as production of CXCL13. It was reported that a high ratio of Tfh cells is strongly correlated with a better prognosis in breast cancer [56,57], and that a reduction in circulating and/or tumor-associated Tfh cells is associated with disease progression and decreased disease-free survival time in hepatocellular carcinoma [58]. According to a former study, Tfh cells control the regulation of germinal center B cells, suggesting that Tfh cells may play key roles in determining the pro- versus anti-tumor effects of intra-tumoral B cells [59]. An ex vivo functional assay demonstrated that a functional Tfh TIL presented signals of an active TLS, which was characterized by humoral and cytotoxic immune responses, and the activated functional Th1-oriented Tfh TIL assisted with immunoglobulin and IFN-γ production [57]. Additionally, the CXCL13/CXCR5 signaling axis is an important component of the TME and is closely related to the formation of TLSs [60]. CXCL13-related gene expression is strongly associated with the presence of TLSs [14] and can serve as a biomarker of TLSs [61]. The up-regulated gene expression of CXCL13 has been proven to be associated with better prognosis in oral cancer, bladder cancer, and ovarian cancer [11,14,61], and CXCL13 was significantly lower in patients with metastatic thyroid cancers, which was associated with an abundance of inefficient Tfh cells [62]. In addition, CXCL13 has an important impact on mobilizing B cells. Paradoxically, CXCL13 has been shown to drive several pro-growth and invasive signaling pathways across multiple tumor types, showing that the CXCL13-B cell axis plays a double role in anti-tumor immunity [63]. IL-17 was also found to be able to promote the formation of TLSs, which were mainly produced by Th-17 cells. It was confirmed that TLSs would reduce in the absence of the IL-17 receptor in transgenic mice. The expression of genes Th-1 and Th-17 was found to increase in TLSs, with better prognoses in patients with pancreatic cancer [26], while Th-17 cells were found to have no relation with prognosis in gastric cancers [51]. However, to achieve a relatively balanced state in TLSs, immune suppressive cells are essential to immune niches. T follicular regulatory cells (Tfr) have also been found in TLSs, with a high expression of CXCR5 and FOXP3. Tfr cells were demonstrated to restrain the generation of antibodies specific to the neoantigen with unique access to the B cell follicles, which limited the overall expansion of antigen-specific B cell clones; this limited expansion also favors the emergence of B cell clones secreting high-affinity antibodies [64]. The activity of tumor-associated TLSs was determined by the relative balance between functional Tfh TILs and functional Tfr TILs, with a higher ratio of Tfh cells associated with better prognosis [57]. According to a former study, the suppression of Tfr cells and Treg cells via immune checkpoint inhibitor anti-CTLA-4 and anti-PD-L1 leads to the overexpression of Tfh, resulting in immune adverse events such as excessive and dysfunctional antibody responses of B cells, leading to autoimmune diseases [65]. In addition to Tfr cells, FOXP3+T regulatory cells (Treg) are also involved in immune suppression processes. FOXP3+T regulatory cells (Treg) are extremely low in TLSs but were found to exist in most TLSs in a genetically engineered mouse model of lung adenocarcinoma [66,67]. In addition to suppressing the anti-tumor immune response through silencing anti-tumor immune surveillance and exhausting T cells [68], Treg cells are able to suppress the formation and function of TLSs [21]. Guang-Yu Ding et al. established a quaternary TLS scoring system, in which Treg cell frequencies were strongly associated with worse survival in intrahepatic cholangiocarcinoma [47].

CTLs as principal executors of anti-tumor immune action were found to be significantly lower in TLS follicles than non-follicle areas [69], but CTL density was positively associated with TLS density. NSCLC patients with a high abundance of CTL infiltration and a high density of TLSs had significantly improved overall survival compared with patients with a high abundance CTL infiltration without TLSs [70]. Previous literature has also reported that the co-existence of tumor-associated CTLs and CD20+ B cells in TLSs was found to be associated with improved survival in melanoma [15]. However, another study showed that Intra-tumoral CXCL13+CD8+ T cell infiltration, which was positively correlated with TLS density, indicated inferior clinical outcomes in clear cell renal cell carcinoma [71]. The better clinical outcomes were found to be significantly associated with high infiltrating CTLs regardless of TLS status, which provides proof for the necessity of T cells in antitumor immunity, although they might finally become exhausted [7].

### 3.2. B Cells

B cells are mostly located in the germinal centers of TLSs in human cancers. They are characterized by different markers depending on their maturation degree such as CD19, CD20, and CD21. B cells and plasma cells (mature B cells) make up the germinal centers in TLSs and are considered one of alternative markers of TLSs. In the beginning of the era of immune therapy, B cells were reported to potentially favor tumor occurrence, progression, and spread [72]. In a variety of mouse models, complement and antibodies produced by plasma cells were found to contribute to chronic inflammation [73], and immune complexes might activate macrophages to produce vascular endothelial growth factors that could increase angiogenesis [74]. In addition, B cells were considered to be able to produce suppressive cytokines such as IL-10, inhibiting T cell responses [75]. With the development of tumor immunity, increasing research has suggested that an abundance of B cells, especially in TLSs, has been positively correlated with prognosis and the efficacy of immune therapy in human cancers in recent years [76,77,78,79,80]. Although the true mechanisms by which B cells in TLSs enhance or directly develop anti-tumor immune responses still need to be explored, we can learn from how B cells influence immunity in SLOs. Just like in SLOs, B cells can recognize neoantigens via B cell receptors and then allow antigen binding with major histocompatibility complex-1 (MHC-1) or major histocompatibility complex-2 (MHC-2), then the presentation to T cells directly or to dendritic cells (DC) to activate T cells in TLSs [81,82,83]. This method of antigen presentation is very effective in eliciting a T cell response with a low tumor mutation load and amplifying an immune response with high tumor mutation load [72] because B cells may make contact with tumor cells at a very close distance, and immune complexes formed by combination of antibodies and neoantigens can be internalized by DCs. This means that the quantity of antigens necessary to induce a T cell response is much lower than direct antigen presentation by DCs. In addition, they are able to produce antibodies that can recognize shared tumor antigens, not patients’ specific tumor antigens that are almost recognized by T cells [84]. Tumor cells are damaged by these antibodies through antibody-dependent cell-mediated cytotoxicity (ADCC) and/or antibody-dependent cell-mediated phagocytosis (ADCP). These reactions are mediated by fragment crystallizable (Fc) portions of tumor-specific antibody binding to Fc receptors of effector cells or complements. Similarly, there are differences between B cells with different functional markers in prognosis, such as OS being longer for TLSs with low fractions of CD21+ B cells, and shorter for those with a low activation-induced deaminase (AID)+ fraction of B cells [85]. B cells will gradually decrease the expression of CD21 and increase the expression of AID in their mature process and migrate to the GCs. AID supports immune system diversification and acts in antigen-stimulated B cells by allowing antigen-driven immune globulin diversification. When AID is activated with appropriate cytokine signals in the B cells, interaction can occur with DCs and Tfr cells in GCs [86]. Many studies have indicated that B cells play a direct or indirect important immune role in TLSs. However, a previous study on hepatocellular carcinoma showed that B cell-rich TLSs constitute a specific niche by which to protect tumor progenitors and produce lymphotoxin β to support the growth of tumor cells [87].

### 3.3. Dendritic Cells

Dendritic cells (DCs) are a diverse group of professional antigen-presenting cells, with key roles in the initiation and regulation of innate and adaptive immune responses [88]. DCs are crucial to TLS formation [89] and maintenance, which has been validated in mouse models [90]. LAMP+ DCs (mature DCs) are considered to be believable markers of TLSs in non-small-cell lung cancer (NSCLC), because they are almost exclusively found in these structures in this cancer type [18]. However, some research on other cancers has shown that LAMP+DC was detected in non-TLS tumor lesions [80,91,92]. Another previous study suggested that the LAMP+DC density was correlated with favorable clinical outcomes (overall, disease-specific, and disease-free survival) and the TIL density (in particular, Th-1 cells) was significantly decreased in tumors poorly infiltrated by LAMP+DCs [45]. LAMP+DCs are also strongly correlated with Th-1 cells and immune cytotoxicity signals, and are positively associated with OS, because they can support TLSs to participate in promoting protective immunity in NSCLC. Another major subtype of follicular dendritic cells is discussed later.

### 3.4. High Endothelial Venules

Tumor-associated HEVs characterized by MECA-79 and peripheral node addressin (PNAd) are frequently found in TLSs and have been proposed to play important roles in lymphocyte entry into tumors, which is a process essential for successful antitumor immunity [93]. In a murine model of colon carcinoma, HEVs were observed to control the formation of TLSs via production of IL-36γ [94]. Numerous studies have shown that the density of HEVs is strongly correlated to the density of TLSs and is a positive predictor in many cancer types [27,94,95,96]. 

### 3.5. TLS-Associated Immune Cells

Immune fibroblasts are considered necessary for the early phase of TLS formation via building a network whose expansion is mediated by IL-22 and lymphotoxin α1β2 (LTα1β2) to support TLSs [97,98]. Some studies have shown that TLSs are not promoted by chronic inflammatory conditions in all organs because immune-associated fibroblasts are necessary and indispensable [99,100]. In a mouse model of TLSs, the subcutaneous injection of immune fibroblasts successfully induced TLSs that attracted the infiltration of host immune-cell subsets [17]. Follicular dendritic cells (FDCs) are a specialized type of DC and are detected in the germinal center via labeling CD21, serving as immune-associated fibroblasts [101]. FDCs form a dense three-dimensional follicular network, which lays a foundation for the generation of TLSs. In addition to antigen presentation and providing structural support, FDCs are able to modulate B cell diversity and enhance B memory cell differentiation in GCs [102,103]. The abundance of FDCs has been positively associated with the density of TLSs, suggesting better prognosis [11]. 

Macrophages are characterized by the expression of CD68 and multiple functions. A previous study suggested that macrophages could secrete IL-36γ to control TLS formation [94] and were responsible for recruiting CD4+ T cells and B cells to promote the formation of TLSs as antigen presentation cells [104]. Additionally, macrophages are one of the main types of effector cells of ADCC and ADCP, which are primary anti-tumor mechanisms of humoral immunity in solid tumors. However, following ADCP, macrophages may up-regulate PD-L1 and indoleamine 2,3-dioxygenase to support local immunosuppression [105]. In addition, a previous study on soft tissue sarcomas showed that macrophage colony-stimulating factor-1 receptor (CSF1R) responses were more frequent in TLSs compared with tumor tissue without TLSs. CSF1R is a marker of immunosuppressive macrophages, which are believed to maintain an anti-inflammatory niche for malignant cell growth [106].

## 4. Predictive Roles of TLSs

As we have mentioned, a TLS is a kind of ectopic lymphoid organ, and it is an important part of the immune microenvironment in terms of composition and location [18,20]. With a deeper understanding of TLSs, research related to the impact of cancer treatment on TLSs has also begun to emerge. We discuss the response and prognostic value of TLSs in existing observational studies according to different treatment methods.

### 4.1. Surgery

Some studies have been based on TLSs in surgical specimens to determine the relationship with favorable outcomes; these are summarized in Table 1. The high number of tumor-infiltrating mature DCs (DC-LAMP) may identify patients with early-stage NSCLC having longer DFS (*p* = 0.0056) [45]. Intra-tumoral TLSs may have reflected the existence of an on-going, effective anti-tumor immunity in two hepatocellular carcinoma cohorts, which was associated with a lower risk of early relapse (*p*_1_ = 0.002, *p*_2_ = 0.006) [46]. In resected G1/G2 pancreatic neuroendocrine tumors, the presence of TLSs is an independent and favorable predictor indicating longer DFS (*p* < 0.001) and OS (*p* = 0.001) [35]. Findings on ovarian carcinoma suggest that the presence of mature DCs (DC-LAMP^+^) in the tumor microenvironment may have represented a prognostic biomarker for two independent cohorts that indicated better DFS (*p*_1_ < 0.0001, *p*_2_ = 0.012) and OS (*p*_1_ = 0.0002, *p*_2_ = 0.003) [44]. Radiotherapy, chemotherapy, and immunotherapy will have a certain impact on the quantity and nature of TLSs [107,108,109,110,111,112]. Such studies reveal that TLSs play a role in resisting tumors in the occurrence and development of tumors and are associated with a favorable prognosis.

### 4.2. Immunotherapy

As an ectopic immune organ, a TLS is inextricably linked to the immune microenvironment and immunotherapy. Although the mechanism of action in immunotherapy is still not fully understood, TLSs also play a significant predictive role in the efficacy of immunotherapy. Recent clinical trials are presented in Table 2.

Helmink, BA et al. investigated the potential predictive role of B cells and TLSs in response to immunotherapy, describing the changes in TLSs related to treatment and their association with efficacy [110]. A new immunotherapy evaluation index called the ‘Immune-Related Pathologic Response Criteria’ includes morphological counting of TLSs. The criteria are significantly different between major pathologic responders and non-responders (*p* < 0.05) and have also been shown to be reproducible amongst pathologists [113]. In tumors of both LC and LC-COPD patients, TLS areas and B cells significantly increased with significantly decreased numbers of T cells (*p*_COPD_ < 0.01, *p*_nonCOPD_ < 0.05). In addition, greater areas of TLSs and proportions of B cells were associated with longer survival rates (*p* = 0.027) [31]. TLSs play a key role in the immune microenvironment in melanoma by conferring distinct T-cell phenotypes, and can prolong the overall survival of patients (*p* = 0.006) [15]. Like the biomarkers of chemotherapy for breast cancer mentioned earlier [115], immunotherapy for melanoma uses the same 12-chemokine gene expression score to identify TLSs, revealing a possible link to the long-term survival of patients [43]. This indicates that 12-chemokine gene expression reflects the essential characteristics of TLSs rather than other changes caused by different treatments. TILs are components of TLSs, and adjacent TLSs are correlated with TILs (r [2] = 0.558, *p* < 0.001) in breast cancer. TILs are associated with significantly longer disease-free survival in patients with HR-tumors (*p* = 0.008), but TLSs show no significant relation either in HR+ (*p* = 0.236) or HR-(*p* = 0.25) subgroups [114]. In another trial on breast cancer, TLSs showed predictive roles in DFS (*p* = 0.0265) but not OS (*p* = 0.0887) [19]. In breast cancer, the status of HR and HER2 has a great impact on the immune microenvironment and treatment strategies, and analysis of different subgroups should be considered in subsequent studies. TLSs are associated with longer OS (>3 y) when Th17 pathway genes are overexpressed, and IL17A plays the most significant role. However, the expression of Treg-related genes is associated with relatively short OS (<1.5 y), indicating that the type of TLS is also closely related to disease prognosis [112]. However, in patients with urothelial carcinoma, the baseline TLS level was lower in CR patients, and the TLS change fold was higher after neoadjuvant immunotherapy, suggesting certain cancer types with less lymphocyte infiltration in the immune microenvironment, which can be activated by immunotherapy [109]. DC and T cells in TLSs contribute to the favorable prognosis of colorectal cancer *p*_DFS_ = 0.0031, *p*_OS_ = 0.006 [32]. In oral squamous cell carcinoma, HEV, a possible symbol of TLS formation, is associated with a favorable immune microenvironment [42], and the presence of TLSs can predict longer DFS (*p* = 0.002) and OS (*p* < 0.001) [30]. In pancreas-related tumors, TLSs [28,40] or a similar distribution of immune cells within TLSs [33] predict survival benefits in patients. The relationship between immunotherapy and TLSs is very complex [115], and its specific mechanisms and long-term efficacy observations need to be further studied.

## 5. Possible Methods in Inducing TLS Formation

### 5.1. Cancer Therapy

Cancer and non-cancer therapies that can induce TLSs is summarized in Table 3.

In mouse models, LIGHT-VTP can induce TLS formation, and a large number of intratumoral effectors and memory T cells are generated with ensuing survival benefits [116].

By comparing biopsy specimens before and after neoadjuvant chemotherapy, it was found that neoadjuvant chemotherapy can induce the formation of TLSs (*p* < 0.001) [107]. However, a clinical trial showed no significant changes in TLS components (CD8+ cell and DC-LAMP^+^ cell) after neoadjuvant chemotherapy for lung cancer [108]. After neoadjuvant chemotherapy, TLSs showed different changes, which may be affected by sample size, cancer type, drugs, treatment cycle, etc. It is necessary to not only observe the relationship between the existence of TLSs and efficacy, but to also study the changes in TLSs before and after treatment.

As for immunotherapy, in one study the TLS density in the CR group was lower than that in the non-CR group before treatment (*p* = 0.074), but there was no statistical difference after treatment (*p* = 0.43). Unsurprisingly, TLS fold change is significantly higher in CR patients compared to non-CR patients (*p* = 0.00039) [109]. Although the density of TLSs has a predictive effect on the treatment efficacy, its change exhibits no significant difference before and after immunotherapy. However, CD20 significantly increased before and after immunotherapy among responders, indicating possible changes in components instead of density [110]. Agonistic CD40 therapy induces TLS formation in mice with glioma but impairs response immunotherapy by suppressing CD11b+ B cells [118]. The research above demonstrates the possibility of adverse effects accompanied with increases in TLS formation, which has been mentioned in the second part of this review.

### 5.2. Non-Cancer Therapy

A 3D-printed scaffold vaccine in mice can induce the formation of TLSs, but its relationship with cancer progression and treatment efficacy remains to be explored [117]. Immune-related non-cancer therapies include vaccination, implantation of artificial objects, etc. In HPV-vaccinated subjects, tissue T cells form TLSs and show characteristics associated with in vivo activation via T cell receptor engagement with cognate antigens [111]. In addition, it has been reported that colony-stimulating factor (GM-CSF)-secreting allogeneic PDAC vaccine is related to the aggregated presence of TLSs (*p* < 0.0001) [112]. 

## 6. Discussion

TLSs, as the main vectors of B cell responses and important helpers of T cell responses, will receive more attention in the era of immunotherapy. The spatial structures of the tumor’s immune infiltrating cell clusters provide complex and cytokine-rich niches in the TME. On the one hand, intratumoral lymphoid structures can develop more powerful and quicker immune responses compared with SLOs; on the other hand, proinflammatory cells and various cytokines in TLSs can protect tumor cell growth and promote metastasis. In the process of summarizing clinical trials, we found that most predictors related to TLSs have a certain prognostic effect, whether it is simply observation of the prognosis of surgical patients or perioperative adjuvant/neoadjuvant therapy. The prognostic value includes short-term efficacy indicators (ORR) and long-term survival indicators (DFS, OS). As part of the immune microenvironment, TLSs hold great promise for efficacy prediction and the enhancement of efficacy in future studies. However, there are still some problems and challenges to be addressed before TLSs are widely admitted and used in clinical practice.

Some studies have suggested that not only the quantity, but also the compositional changes in TLSs, can play a predictive role [50,110]; however, such prognostic factors have not been fully studied. A large number of studies mainly explore the value of TLSs in a broad sense, but the content related to TLS component analysis is very limited. As mentioned in the first part of the article, in TLSs, there are some cells that can promote the body’s immune response, such as Th1 cells, B cells, and LAMP+DCs; some have an inhibitory effect on the immune response, which will promote the growth and escape of cancer cells, such as Th2 cells, Treg cells, and macrophages. If some factors can be considered comprehensively in order to establish a classification or scoring standard, this may play a positive role in the prognosis evaluation of patients [119]. However, in order to study different types of TLS, a unified identification standard should first be developed; that is, what kind of lymphocyte composition and spatial distribution can be considered to be a TLS? At present, it is generally believed that the basic structure of a TLS comprises peripheral T cells, central B cells, mature DCs, and HEVs [18,20]. The identification of TLSs components mainly relies on immunohistochemistry, and the main markers are described in Figure 2. The morphological resolution mainly relies on the appearance of H&E staining, immunohistochemistry and immunofluorescence for manually determination [119]. Flow cytometry and gene sequencing are also used to better identify TLSs. 

It is worth emphasizing that TLS is a heterogeneous spatial structure. Its location, density, and components are influenced by different cancer types as well as host immune function. At present, there is only a small amount of basic research on the signaling of immune cells and immune-related cells in the process of TLS formation. It is necessary to study the interactions of immune cells, immune related cells, and stromal cells in TLSs in the future, which could help the clinicians and researchers better understand the role and mechanisms of TLSs in tumors and to propose possible targeted therapeutic approaches.

The number of clinical trials on TLSs has increased to a certain extent, including four trials of observations after surgery alone, 11 trials assessing chemotherapy efficacy, and 15 trials assessing immunotherapy efficacy. However, cancer types are relatively scattered. Although some studies have shown relatively significant predictive effects, they lack the value of mutual comparison and validation. In addition, in respect of breast cancer, the predictive role of TLSs was only manifested in the HR-subgroup [114], suggesting that we need to consider gene mutation status in future studies because its impact on the tumor microenvironment may also be involved in TLS formation and functioning. In addition, some studies of neoadjuvant therapy are designed for TLSs in chemotherapy [27,51,56,95,108,120], but neoadjuvant immunotherapy was not involved with some studies, which excluded patients receiving neoadjuvant therapy [19]. With the wide application and promising efficacy of neoadjuvant immunotherapy [121], more patients are able to undergo surgical resection, which suggests a certain convenience for the assessment of the patient’s immune microenvironment and TLS conditions.

The induction of TLS formation to enhance efficacy is still in the early stages; although changes in TLS formation and composition have been observed following immunotherapy and chemotherapy [107,108,109,110], there are also non-cancer therapies that can induce TLS formation [111,112]. In the current stage, the active induction of TLSs is still only applied in mouse models and cannot modulate the composition of TLSs; as a result, there is room for improvement in research in this area. In addition, the changes in TLSs before and after treatment should be actively observed, and long-term follow-up should be undertaken for patients who receive tumor-preventive vaccines. Attention can also be given to patients with tumors implanted with artificial objects, such as patients with arterial and bile-duct stents.

## 7. Conclusions

TLSs have received increasing attention from researchers and clinicians as a specific component in the immune microenvironment. As the composition and biological behavior of TLSs have become clearer, TLSs have been included in more and more studies. Furthermore, research on changes in the location, density, and composition of TLSs and their effects are also being conducted on a small scale, exploring the future direction of TLSs research. However, there is no standardized identification method for TLSs, and the number of studies for each specific type of tumor is still insufficient. As a result, the role of TLSs in tumor treatment selection and prognosis estimation remains unclear. Overall, we can believe that with continuous research on TLSs, it can provide help for clinical treatment and prognosis assessment. If the role and mechanisms of TLSs are confirmed, it will hopefully become a routine test for pathological diagnosis and widely used in clinical practice.

## Figures and Tables

**Figure 1 cancers-14-05968-f001:**
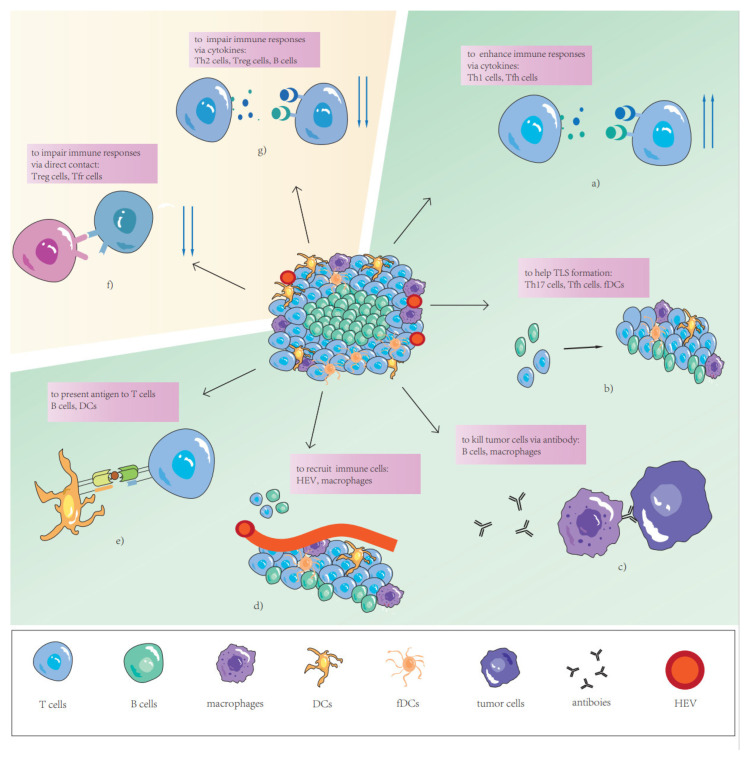
The role of different TLS components. (**a**) Th1 cells and Tfh cells produce cytokines to enhance immune responses. (**b**) Th17 cells, Tfh cells, and fDCs help TLS formation via cytokines or contact. (**c**) Plasma cells kill tumor cells via antibody-dependent cell-mediated phagocytosis (ADCP) and or antibody-dependent cell-mediated cytotoxicity (ADCC). (**d**) Circulating immune cells migrate into TLSs via HEVs or macrophage induction. (**e**) B cells and DCs present antigens to T cells. (**f**) Treg cells and Tfr cells impair immune responses via direct contact. (**g**) Th2 cells, Treg cells and B cells produce cytokines to impair immune responses.

**Figure 2 cancers-14-05968-f002:**
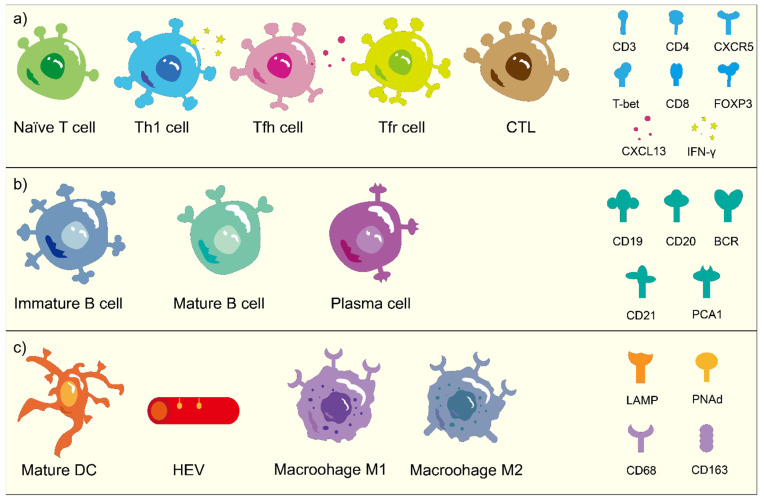
The cellular biomarkers of TLS. (**a**) The subtypes of T cells and cellular markers. (**b**) The different stages of B cells and cellular markers. (**c**) The TLS-associated cells and cellular markers.

**Table 1 cancers-14-05968-t001:** Predictive roles of TLSs in patients treated only with surgery.

Tumor	Number of Patients	Identification Method	Predictive Roles
lung cancer [45]	74	H&E staining, immunohistochemistry	DFS
hepatocellular carcinoma [46]	273	H&E staining, immunohistochemistry	early relapse
pancreatic neuroendocrine tumor [35]	182	H&E staining, immunohistochemistry, immunofluorescence	DFS OS
ovarian carcinoma [44]	167	H&E staining, immunohistochemistry, RNA-seq, flow cytometry	OS

**Table 2 cancers-14-05968-t002:** Predictive roles of TLSs in patients treated with immunotherapy.

Tumor	Number of Patients	Identification Method	Predictive Roles
solid tumor [7]	328	H&E staining, immunohistochemistry, immunofluorescence	DFS OS
melanoma renal cell carcinoma [110]	melanoma 20	H&E staining, immunohistochemistry, immunofluorescence, RNA-seq, flow cytometry	ORR
	renal cell carcinoma 14		
lung cancer [113]	20	H&E staining	ORR
lung cancer [31]	133	H&E staining, immunohistochemistry	OS
melanoma [15]	177	H&E staining, immunohistochemistry, RNA-seq, immunofluorescence	OS
melanoma [43]	120	H&E staining, immunohistochemistry, RNA-seq	ORR
breast cancer [114]	447	H&E staining, immunohistochemistry	DFS (HR-)
breast cancer [19]	167	H&E staining, immunohistochemistry	DFS
pancreatic cancer [112]	39	H&E staining, immunohistochemistry, immunofluorescence	OS
colorectal cancer [32]	149	H&E staining, Immunohistochemistry, RNA-seq	DFS OS
oral squamous cell carcinoma [42]	75	H&E staining, immunohistochemistry, immunofluorescence	DFS
oral squamous cell carcinoma [30]	168	H&E staining, immunohistochemistry, RNA-seq, immunofluorescence, flow cytometry	DFS OS
pancreatic cancer [28]	63	H&E staining, immunohistochemistry, RNA-seq, flow cytometry	OS
pancreatic cancer [33]	104	H&E staining, immunohistochemistry, RNA-seq	OS
pancreatic ductal adenocarcinoma [40]	51	H&E staining, immunohistochemistry	OS

**Table 3 cancers-14-05968-t003:** Cancer and non-cancer therapies inducing TLSs.

Tumor	Experimental Subject	Treatment	Identification Method	TLS Change
resistant cancer [116]	mice	LIGHT-VTP immunotherapy	H&E staining, immunohistochemistry, immunofluorescence, flow cytometry	increase
cancer [117]	mice	3D printing scaffold vaccine	H&E staining, flow cytometry	increase
glioma [118]	mice	agonistic CD40 therapy	H&E staining, immunohistochemistry, immunofluorescence, flow cytometry, RNA-seq	increase
hepatoblastoma [107]	12 patients	neoadjuvant chemotherapy	H&E staining, Immunohistochemistry, RNA-seq	increase
lung cancer [108]	122 patients	neoadjuvant chemotherapy	H&E staining, immunohistochemistry	not significant
urothelial cancer [109]	24 patients	neoadjuvant immunotherapy	H&E staining, immunohistochemistry, immunofluorescence, RNA-seq	increase
melanoma renal cell carcinoma [110]	34 patients	maintenance immunotherapy	H&E staining, immunohistochemistry, immunofluorescence, RNA-seq, flow cytometry	not significant
cervical [111]	12 patients	HPV 16 vaccine	H&E staining, Immunohistochemistry, RNA-seq, flow cytometry	increase
pancreatic cancer [112]	39 patients	pancreatic tumor vaccine	H&E staining, immunohistochemistry, RNA-seq	increase

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
