# Peer review of "Tertiary Lymphoid Structures: A Potential Biomarker for Anti-Cancer Therapy"

_cancers, 2022, doi:10.3390/cancers14235968_

Round 1

Reviewer 1 Report

This is a good review of Tertiary Lymphoid structures (TLS) and their role as potential biomarkers for cancers.Strengths:

- The review details TLS components and their role in tumor-specific immune response, possible methods to induce TLS formation and their predictive role in patients treated with surgery, immunotherapy and chemotherapy.

- The references are most recent and comprehensive.   
- Interpretation and presentation of previous studies is accurate.              
- No major suggestions for improvements.          
- Clarity and context in this paper are good.

Minor comments:           
- The figure 1 could use a better textual font as it is currently too small and pixelated.

- Language editing and proper punctuation is needed.

Author Response

Dear reviewer 1:
Thanks for your comments, here are our responses and revision in detail.

We have reuploaded figure 1 with higher quality, and we have also improved the image quality when uploading newly added figure 2. The image quality of the figure inserted into word document will be compressed, please see the pdf we uploaded as an attachment.

We have used a paid editing service from MDPI for our manuscript to undergo extensive English revisions. And the conformation will be attached when we submit revised manuscript.

Reviewer 2 Report

This manuscript by Zou et al. reviews on tertiary lymphoid structures (TLS) as a potential marker for anti-cancer therapy. Authors provided discussion on key aspects of the function of TLS. I think this review covers useful information on the role of TLS components (such as T cells, B cells, DCs, TLS-associated cells) in tumor related immune response. I think this review will be very interesting to the readers of Cancer. However, there are some issues need to be addressed before accepting the manuscript.

-  Authors have provided the schematic representation of the functional aspects of different components of TLS in Figure 1. However, the structural models are not provided in a clear way. I would strongly suggest authors providing three dimensional structural models of TLS components that can be helpful for readers to understand structure-function relationships more clearly.

-Authors have provided discussion on the perspective of the role of TLS in section 6.Discussion. But the conclusion/summary of this review is missing in the manuscript. Authors need to provide conclusion/ summary of this review article.

-There are some typos, such as: page 2, line 67, “…able to provides local...” should be “…able to provide local...”;  page 2, line 69, “At present TLS could..” should be “At presentTLS could…”; page 7, Figure1 label, line  310, “d circulating…” should be “d Circulating…”

Manuscript needs a proof read.

Author Response

Dear reviewer 2:
Thanks for your comments, here are our responses and revision in detail.  

Currently, the structure of TLS is not yet completely understood. Pathological examination is only possible to see the structure in planes, and spatial technology is relatively little used in the identification and study of TLS, so there is some difficulty in representing it graphically in three dimensions. We will adjust the graphic structure and arrangement, change the color matching and presentation, etc., to try to show the composition and structure of TLS as demonstrated by existing studies.

We have added a 7.Conclusion part after section 6.Discussion (page 15, line 547-559, please view without showing correlations), which generally summarizes the present research status, existing problems and prospective of TLS.

We have used a paid editing service from MDPI for our manuscript to undergo extensive English revisions. And the conformation will be attached when we submit revised manuscript. Also, we have proofread ourselves to check for possible residual spelling mistakes.

Reviewer 3 Report

In their manuscript, Zou et al. present the importance of TLS in immunotherapy. Further, the authors highlight the components of the TLS and summarize the observations made in the TLS before and after the immunotherapy in different cancer patients. The authors also mentioned the different methods of TLS induction in their study. This manuscript presents evidence that collectively suggests the importance of TLS in predicting of cancer immunotherapy and highlights the need for more studies in this field. Overall, the authors have highlighted a hitherto underappreciated topic using convincing references. However, a few concerns need to be addressed.

The writing is sloppy and requires major language editing.

The authors should also add agonistic CD40 therapy in inducing tertiary lymphoid structures in glioma. PMID: 34226552

The authors must include a separate section to discuss the adverse effects of TLS in cancer.

Author Response

Dear reviewer 3:
Thanks for your comments, here are our responses and revision in detail.

We have used a paid editing service from MDPI for our manuscript to undergo extensive English revisions. And the conformation will be attached when we are submitting revised manuscript.

We have added agonistic CD40 therapy in inducing tertiary lymphoid structures in glioma (PMID: 34226552) in the section of TLS induction (Table 4 AND page 13, line 470-473, please view without showing correlations). And we have also discussed its negative impact on immunotherapy when we are describing the role of TLSs as a whole (page 3, line 124-134, please view without showing correlations).

We have added a separate paragraph to discuss the adverse effects of TLS in cancer (page 3, line 124-134, please view without showing correlations), which is described separately from the aspects of basic and clinical medicine.

Round 2

Reviewer 2 Report

The revised version of the manuscript is much improved. The authors have addressed the majority of my comments. 

I would like to recommend this manuscript for publication. 

Author Response

Dear Reviewer 2,

Thank you for your great efforts in the review process and we will do our best to improve the quality of this review.

Yours sincerely,

Fang Wu